# Relationship between Microflora Changes and Mammary Lipid Metabolism in Dairy Cows with Mastitis

**DOI:** 10.3390/ani13172773

**Published:** 2023-08-31

**Authors:** Yang Luo, Zhiwei Kong, Bin Yang, Fang He, Cheng Huan, Jianbo Li, Kangle Yi

**Affiliations:** 1College of Animal Science and Technology, Guangxi University, Nanning 530004, China; xinhelu509@163.com; 2Guangxi Key Laboratory of Animal Breeding, Disease Control and Prevention, Nanning 530004, China; 3School of Biological and Chemical Engineering, Zhejiang University of Science and Technology, Hangzhou 310023, China; 4Hunan Institute of Animal and Veterinary Science, Changsha 410131, China; fanghexuke2023@163.com (F.H.); chenghuanxuke2023@163.com (C.H.); ljbljb12@126.com (J.L.); yikangle@yeah.net (K.Y.)

**Keywords:** mastitis, dairy cow, microbial diversity, lipid metabolism

## Abstract

**Simple Summary:**

Mastitis is recognized as the most prevalent dairy cow disease in the world, which directly leads to inflammatory damage of dairy cow mammary tissue, a decrease in mammary gland cell secretion function, and an increase in somatic cell count, thus affecting milk production and quality, and resulting in huge economic losses in the dairy cow breeding industry. Previous researchers have explored the pathogenesis and metabolism of dairy cows with mastitis. However, information on the relationship between microorganisms and lipids in mastitis is limited. This study attempted to explore the relationship between the microorganisms and lipids of dairy cows with mastitis using a 16S rDNA sequencing technology and lipidomics technique. Based on the obtained results, we found that microorganisms caused abnormal lipid metabolism through positive or negative regulation of lipid metabolites, and ultimately decreased milk quality.

**Abstract:**

Dairy mastitis is an inflammatory reaction caused by mechanical injury and stress within the mammary gland, during which microbial changes and abnormal lipid metabolism occur. However, the underlying mechanism is still unclear. The present study used a combination of 16S rDNA sequencing technology and lipidomics techniques to reveal the effects of mastitis on lactic microbiota and metabolites in the milk of dairy cows. Twenty multiparous Holstein dairy cows (2–3 parities) with an average body weight of 580 ± 30 kg were selected for this study. The dairy cows were allocated to control group (<5 × 10^4^ cells /mL)) and mastitis group (>5 × 10^6^ cells /mL) based on the somatic cell count. The results showed that mastitis caused a decrease trend in milk production (*p* = 0.058). The results of the 16 s sequencing indicated a significant decrease (*p <* 0.05) in the number of *Proteobacteria*, *Tenericutes* colonized in mastitis milk, and the number of *Firmicutes*, *Bacteroidetes* and *Actinobacteria* communities increased significantly (*p <* 0.05). The lipidomics results revealed that the changes in lipid content in mastitis milk were correlated with arachidonic acid metabolism, α -linolenic acid metabolism and glycerol phospholipid metabolism. The results showed that mastitis may cause abnormal lipid metabolism in milk by regulating the diversity of milk microflora, and ultimately affect the milk quality.

## 1. Introduction

Mastitis is the most common disease that affects the health conditions of modern dairy farming, and it still exhibits a high incidence in large-scale and intensive dairy farms. It is generally believed that cow mastitis is inflammation caused by mammary mechanical damage and stress caused by milking operations, resulting in pathogenic microbial infection in mammary tissue [1]. Classical microbial pure culture and molecular biological techniques are often used to isolate and identify pathogenic bacteria of mastitis [2]. At present, *Streptococcus* [3], *Escherichia coli* [4], *Staphylococcus aureus* [5] and other major pathogenic bacteria have been isolated from milk. Additionally, *Streptococcus* gapC protein [6] and the Fnbpa adhesion gene of *Staphylococcus aureus* [7] and virulence genes of *Escherichia coli f17c-A* and *f17b-A* [8] provide biological materials for research into the microbial infection mechanism of cow mastitis. However, the structure of microbial flora in mastitis milk is more complex [2], and the traditional pure culture method and molecular biology PCR identification have limitations and ambiguous directivity. Therefore, there is still a risk of blind administration in clinical diagnosis.

Milk fat is an important component of milk, with a proportion of about 3–5%, and is the main energy donor of milk. Its main form is triacylglycerols [9]. In addition, it also includes glycerol (GL), glycerophospholipid (GP), sphingolipid (SP), fatty acid (FA), etc. [10]. These lipids determine the physical and chemical properties of milk fat and the quality of dairy products. There are many lipids in milk with complex structures, and more than 400 lipids have been identified so far, but there are still many lipids with low abundance that have not been identified [11]. The lipid composition in milk is closely related to the living environment of dairy cows, and some studies believe that the lipid content in milk is correlated with the stress state of dairy cows [12,13]. In addition, Thomas′s study found that the phosphoric choline and Sn-glycero-3-clicocholine involved in glycerol adipogenesis were significantly reduced in clinical mastitis milk [14]. At present, the mechanism of the changes in lipid metabolism in milk caused by mastitis is not clear.

At present, there are few studies on the causes of mastitis affecting the diversity of flora and the changes of lipid metabolism in milk [15,16]. Therefore, this experiment uses a combination of 16S rDNA sequencing technology and lipidomics to explore the specific mechanism of mastitis affecting milk quality, which will provide a new target for the follow-up prevention and treatment of mastitis and provide a theoretical basis for ensuring the quality of milk.

## 2. Materials and Methods

This experiment was approved by the Animal Care and Use Guidelines of the Animal Care Committee, Hunan Institute of Animal and Veterinary Science, Changsha, China (Approval Code: 202206). Approval Date: 8 March 2022.

### 2.1. Animals and Experimental Design

According to the Dairy Herd Improvement (DHI) data of a large-scale dairy farm in the Hunan Dairy Production Performance Test Center in June, twenty multiparous dairy cows (580 ± 30 kg, 2–3 years old) were selected based on the milk somatic cells count (SCC). Among them, ten dairy cows were regarded as the control group (Con) with SCC less than 5 × 104 cells/mL, and the other ten dairy cows were regarded as the mastitis group (Mas), with SCC greater than 5 × 106 cells /mL. The basal diet (shown in Table 1) [17] was formulated to meet the nutrient requirements for energy, protein, minerals, and vitamins according to the Feeding Standards of Dairy Cattle in China [18]. All dairy cows were fed the same total mixed ration (TMR) diets ad libitum. All dairy cows had ad libitum access to water.

### 2.2. Sample Collection

For the milk quality analysis, milk samples of dairy cows within Con and Mas used for DHI determination in June were used. In brief, a 50 mL milk sample was collected from each dairy cow. The samples were mixed in the morning and evening according to the ratio of 3:2. Samples were kept at 4 °C for milk quality analysis within 6 h.

For microbial diversity detection and targeted lipidomics analysis, a CMT rapid detection kit (M7510, Invitrogen, NM, CA, USA) was used to confirm the mastitis milk quarter of Mas cows. Then, 50 mL milk samples from one confirmed milk quarter of Mas cows or the right hind quarter of Con cows were collected [19]. The collected milk samples were quickly frozen in liquid nitrogen.

### 2.3. Determination of Milk Quality

The samples were mixed in the morning and evening according to the ratio of 3:2, and then the milk fat rate, milk protein, lactose, milk solid and somatic cell number of milk were measured using an automatic milk composition analyzer (Foss Inc., Hillerod, Denmark) within 6 h.

### 2.4. DNA Extraction and 16S rDNA Sequences

In total, 10 mL of each milk sample was concentrated for 12 min at 3000× *g* (4 °C). The precipitates were washed 3 times in a 0.9% NaCl solution. Bacterial DNA was extracted according to the manufacturer′s instructions (QIAamp DNA Stool Mini Kit, Qiagen, Valencia, CA, USA). The quality of metagenomics DNA was confirmed via 1% agarose gel electrophoresis and diluted to 1 ng/μL as the template for PCR. The V3 and V4 regions of 16S rDNA genes were amplified using the universal forward primer 338F (5′-ACTCCTACGGGAGGCAGCA-3′) and 806R (5′-GGACTACHVGGGTWTCTAAT-3′) with a set of 6-nucleotide barcodes [20]. The amplified products were quantified using a spectrophotometer, and then the DNA was sequenced using a Illumina MiSeq sequencing system.

### 2.5. Bioinformatics Analysis

After demultiplexing, the paired-end sequencing reads were processed using the DA-DA2 package (version 1.8.0) in R and its pipeline. Barcodes and primers were removed, and reads with more than 2 expected errors (maxN = 0, maxEE = c(2, 2), truncQ = 2) were filtered out. After merging the paired reads and filtering the chimera, an ASV table was constructed. The ASVs were assigned to Greengenes database (Greengenes 13.8) using a naive Bayesian classifier method implemented in DADA2 [21]. The Alpha diversity level of each sample was evaluated. The differences and significance of beta diversity between two groups were determined via analysis of similarity (ANOSIM).

### 2.6. Non-Targeted Lipidomics Analysis

Lipidomics is based on the LC-MS/MS system combined quadrupole Orbitrap Mass Spectrometer (Q Exactive Orbitrap, Thermo Fisher Scientific, Waltham, MA, USA). The brief steps are as follows: firstly, take a 100 μL milk sample in a 2 mL tube and add 750 μL chloroform methanol mixed solution (2:1, −20 ℃), vortex oscillation for 30 s. Then, place the mixture on ice for 40 min and add 190 μL ddH_2_O, vortex oscillation for 30 s, standing on ice for 10 min, centrifuging at room temperature of 12,000 rpm for 5 min, and taking 300 μL of lower-layer liquid. Transfer the collected liquid to a new 2 mL centrifuge tube. Then, add another 500 μL chloroform methanol mixed solution (2:1, −20 ℃), and perform vortex oscillation for 30 s. Centrifuge the mixture at room temperature of 12000 rpm for 5 min, and take the lower liquid 400 μL. Transfer the liquid to a 2 mL centrifuge tube and concentrate the sample with a vacuum centrifuge concentrator. Use 200 μL isopropanol dissolved sample, 0.22 μM membrane filtration, to obtain the sample to be tested for LC-MS online detection. Take 20 μL samples from each sample to test the quality, and use the remaining samples to test for LC-MS detection. Lipid search software (V4) [22] was used to annotate the original data to obtain data matrices such as the mass to charge ratio (*m*/*z*), retention time (RT) and peak response value (intensity), and then all data were corrected and normalized. The samples were grouped and analyzed via principal component analysis (PCA), partial least squares discriminant analysis (PLS-DA) and orthogonal partial least squares discriminant analysis (OPLS-DA). Taking the lipid with VIP value >1 as the screening standard for differential metabolites, univariate statistical analysis Student′s *t*-test was performed between groups, and the data with *p* < 0.05 were screened to obtain differentiated lipids.

The data analysis methods for lipidomics are as follows: data analysis is mainly divided into basic data analysis and optional data analysis. Basic data analysis includes Student′s *t*-test, principal component analysis (PCA) and orthogonal projections to latent structures–discriminant analysis (OPLS-DA); alternative data analyses include hierarchical cluster analysis of differential metabolites, radar map analysis of differential metabolites, correlation analysis of differential metabolites, KEGG annotation of differential metabolites, pathway analysis of differential metabolites and regulatory network analysis of differential metabolites.

### 2.7. Statistical Analysis

Statistical analyses of the experimental data were carried out using the SPSS version 23 (SPSS Inc., Chicago, IL, USA). The milk quality parameters were subjected to an independent sample t-test method. The difference between the two groups was evaluated, and differences were considered statistically significant at *p* < 0.05, and trends were recognized 0.05 < *p* < 0.1. Means ± standard errors of the mean (SEMs) were used to present the results.

## 3. Results

### 3.1. The Effects on Milk Performance

The results of the effects of mastitis on milk performance of Holstein dairy cows are shown in Table 2. Compared with the normal group, mastitis could significantly improve the somatic cell number and reduce milk yield by 16.61% and milk fat level by 5.88%. In addition, milk yield had a downward trend (*p* = 0.058). There was no significant difference in lactose, total solids and urea contents.

### 3.2. The Results of Milk-Associated Microbiome

In total, 1,262,294 valid reads were retrieved from milk samples with an average of 8415.29 sequences per sample after quality trimming and chimera checking. The minimum and maximum nucleotides lengths were 175 and 442, respectively, and the two groups shared 1640 OTUs. The pattern of the rarefaction curve confirmed that the sequencing data coverage was adequate to describe the milk-associated bacterial composition in the present study (Appendix A). The Chao1 and Shannon indices were used to assess the alpha diversity of milk-associated microbial profiles. The alpha diversity index was used for statistical analysis from the seven aspects of Chao1, Goods coverage, Simpson, Pielou-e, Faith PD, Shannon and Observed species. It can be seen from Figure 1 that the indexes of Chao1, Simpson, Pielou-e, Faith PD, Shannon and Observed species in mastitis group were greater (*p* < 0.05 or 0.05 < *p* < 0.1) than those in the control group, indicating that the microbial diversity and abundance in mastitis cow milk were higher than those in normal cow milk. Through PCoA analysis (shown in Figure 2), we found that the mastitis group formed separate clusters from the control group (ANOSIM-R = 0.12, *p* = 0.032), indicating that there were great differences in microbial communities between the two groups, and there have been great differences in the microbial flora of dairy cows suffering from mastitis.

The relative abundances of milk-associated microbiota at the phylum level are shown in Figure 3A. There were 10 dominants of bacterial phyla identified with a mean relative abundance of ≥1%. *Proteobacteria* was the dominant colony in normal milk (81.96%), followed by *Firmicutes* (11.14%). The number of *Proteobacteria* caused by mastitis decreased significantly (*p* < 0.05), and *Tenericutes* were colonized in mastitis milk only (15.35%). In addition, the number of *Firmicutes, Bacteroidetes* and *Actinobacteria* communities increased (*p* < 0.05) significantly in the mastitis group, by 81.86%, 78.17% and 108.17%, respectively (Figure 3B–E). The LDA with LEfSe analysis was used to explore microbiota differences from phylum to genus between the two groups (Figure 4). The relative abundance of *Proteobacteria*, *Betaproteobacteria* and *Burkholderiales* was enriched in the control group compared with the mastitis group, while *Tenericutes*, *Mollicutes*, *Mycoplasmataceae* and *Mycoplasmatales* were enriched in the mastitis group.

### 3.3. Non-Targeted Metabolome Profiles of the Milk

The total extracts were subjected to UPLC-TQMS for non-targeted metabolomics to explore the effects of mastitis on the metabolite profiles in the milk of dairy cows. The repeatability of each samples extract was assessed via an overlying analysis of the total ion current (TIC) in the quality control (QC) samples in negative and positive modes. The peaks obtained from all the experimental samples and three QC samples were preprocessed and then subjected to PCA analysis (Appendix A). It was found that the samples basically coincided in the figure, indicating good experimental repeatability, good data quality control, and the reliability of the analysis method used in this study. A total of 374 lipids from 11 subclasses were identified (Appendix A), and 37 significantly different lipids were screened. This included 5 kinds of triglyceride (TG), 12 kinds of phosphatidyl choline (PC), 8 kinds of phosphatidyl ethanolamine (PE), 3 kinds of sphingomyelin (SM), 1 kind of phosphatidyl inositol 1 (PI), phosphatidylserine (PS), 2 kinds of double acid glyceride (DG), 2 kinds of dihexose ceramide (Hex2cer), 2 kinds of lysophosphatidylcholine (LPC) and 1 kind of lysophosphatidylethanolamine (LPE). Volcano plots were used to reveal differential metabolites, and 23 of 37 were upregulated, while 14 of 37 were downregulated in the mastitis group compared with the control group (Figure 5B).

After KEGG metabolic pathway enrichment analysis (Figure 6A), the differentially metabolized lipids detected were involved in three metabolic pathways, including arachidonic acid metabolism, alpha-linolenic acid metabolism and glycerophospholipid metabolism. Arachidonic acid metabolism was the most correlated, followed by linolenic acid metabolism and glycerophospholipid metabolism. The Pheatmap program package in R language was used to scale the dataset, and the hierarchical clustering analysis diagram of the relative quantitative values of 37 differentially metabolized lipids was obtained (Figure 6B). The control group and the mastitis Group could be well clustered together separately, indicating that there were no significant differences within the sample groups and there were significant differences between the groups.

### 3.4. Correlation Analysis among the Milk Qualities, Microbiome and Lipidomics

The production performances, differential metabolites and top 50 bacterial genera were used for Spearman correlation analysis (Figure 7). Somatic cells are positively correlated with *Bifidobacterium*, *Ureaplasma*, *Atopococcus* and PS, TG and PC (*p* < 0.05), while negatively correlated with LPE, PE and TG 1 (*p* < 0.05). *Ureaplasma* is positively correlated with somatic cell count, TG, PI and LPC (*p* < 0.05), but negatively correlated with PE (*p* < 0.05). PE was positively correlated with *Mycoplasma, Psychrobacter* and *Acholeplasma* (*p* < 0.05), and negatively correlated with milk production (*p* < 0.05). PC was correlated with *Acholeplasma, Ruminobacter, Trepomemay, Bifidobacterium* and somatic cell count (*p* < 0.05).

## 4. Discussion

Cow mastitis is the most common disease on dairy farms. According to the National Mastitis Committee (NMC) of the United States, there are about 220 million dairy cows in the world, and about 30% of them are suffering from mastitis to various degrees. In the United States, the United Kingdom and Japan, the incidence of mastitis in dairy cows is as high as 45%, 40–50% and 45.1%, respectively [23]. Research shows that when dairy cow mastitis influences lactation function, due to the secretion of mammary tissue damage, mastitis will reduce milk yield [24], which is consistent with the results of this experiment. This is probably because when mastitis occurs, in order to eliminate the pathogen and repair the tissue damage, the body produces a large number of white blood cells. In this scenario, the milk cannot be excreted normally, resulting in a reduction in the total amount of some lactating cells [25]. This may also lead to a decline in milk production caused by the severity of mastitis infection, the type of bacteria, the time of infection and other variation factors [26]. This experiment also determined that the content of lactose in milk was decreased, but not significant, which was not consistent with previous descriptions [27]; the specific reasons for this need to be further explored. The cell count (SCC) in milk is the main indicator for detection and diagnosis of mastitis. Studies have shown that cows with high SCC have reduced milk yield [28], which is consistent with the results of this study to a certain extent, possibly resulting from the changes in the abundance of microbiota. [29].

The probability of a cow suffering from mastitis depends on the age, breed, immune status, lactation stage [30], environmental factors [25] and even the milking equipment used to milk the cow [31]. Among these causes, the most recognized cause of mastitis is the invasion of pathogens [32]. But the exact mechanism is still uncertain. Therefore, 16SRNA sequencing was carried out in this experiment to explore its specific mechanism of action. Studies have shown that the OTU diversity of microbial populations in healthy milk, colostrum and nipple is much greater than that of dairy cows with mastitis [33]. However, in this study, the OTU/ASV of microbial populations in the Mas Group was much greater than that in the Con group, which might be due to differences in sample preparation and storage, methods of DNA extraction, and reading length and depth [34]. In addition, previous studies also showed that there were great differences between the microbial population OTU in the milk of healthy dairy cows and dairy cows with mastitis [35]. Therefore, more studies are needed to confirm the correlation between the abundance of microbial population OTU and the prevalence of mastitis.

At the phylum level, *Firmicutes*, *Proteobacteria*, *Bacteroidetes* and *Actinobacteria* were identified as the core structural microbial flora in healthy cow milk [36], which is consistent with the results of this study. Surprisingly, in this study, it was also found that the number of *Proteobacteria* was significantly reduced, while that of *Firmicutes*, *Bacteroidetes* and *Actinobacteria* was significantly increased, which might be because the invasion of pathogenic microorganisms would break the balance of the original health state and change the proportion of the original microbial population [37]. Additionally, we found that *Tenericutes* were colonized in mastitis milk. Studies have confirmed that *Tenericutes* were related to colitis in the digestive tract, and the authors observed an increase in *Tenericutes* (that live in the environment) because *Mycoplasma* spp. (from *Tenericutes*) was responsible for mastitis [16]. This is also consistent with the descriptions in our study, which indicate that there is a certain correlation between microorganisms in the digestive tract and the occurrence of dairy cow mastitis.

At the genus level, the genera *Ruminococcaceae*, *Lachnospiraceae*, *Propionibacterium*, *Stenotrophomonas*, *Corynebacterium*, *Pseudomonas*, *Streptococcus*, *Comamonas*, *Bacteroides*, *Enterococcus*, *Lactobacillus* and *Fusobacterium* are microorganisms commonly distributed in the milk of healthy dairy cows [38]. In this study, it was found that *Corynebacterium* was relatively abundant in the mastitis group, which was consistent with the results of previous studies stating that *Corynebacterium* could cause mastitis [39]. This may be because *Corynebacterium* has a significant correlation with the number of somatic cells in liquid milk [40]. *Streptococcus*, *Staphylococcus aureus*, *Escherichia coli* and *Bacillus* are recognized as the representative pathogenic bacteria causing cow mastitis worldwide. However, the distribution of pathogenic bacteria in the affected areas is different [41], which also causes an imbalance in the proportion of structural microbial flora in milk in healthy conditions [42]. The results of this study showed that the abundance of *Streptococcus* increased in the mastitis group, while the abundance of *Bacillus* decreased, which may be because the somatic cell number of cows was positively correlated with the abundance of *Streptococcus* but negatively correlated with the abundance of *Bacillus* [43].

In addition, in the top 20 microbial abundance levels of mastitis samples collected in this study, *Staphylococcus* was found in seven samples, *Mycoplasma* was found in six samples, and *Ureaplasma* was found in five samples. Due to the particularity of its growth medium, mycoplasma is often ignored in laboratory culture identification and drug sensitivity tests [44]. Among more than 200 mycoplasma pathogens found so far, 5.5% are confirmed to be related to dairy mastitis [45]. Moreover, mastitis caused by mycoplasma has strong resistance to antibiotics [46], which is consistent with our previous clinical practice results. In conclusion, it can be concluded that mastitis in this study is likely to be caused by the cross-infection of *Staphylococcus aureus*, *Mycoplasma* and *Ureaplasma*, which is consistent with the results described previously [47].

With the continuous improvement and perfection of metabonomic analytical instruments and analytical techniques, more and more studies have used metabonomics to analyze cow milk samples to explore the physiological function and lactation mechanism of the mammary gland [15,48]. In the present experiment, we found that mastitis could cause a significant decrease in the milk fat level of dairy cows, which was consistent with the research results of Wang [49]. This may be due to the increase in somatic cells caused by various factors [50], resulting in a decrease in the milk fat level. Although there is some research on the mechanism [51], the specific role of lipid metabolism and its relationship with mastitis has not been clarified. Therefore, this experiment uses lipidomics technology to deeply explore the role of lipid metabolism in mastitis adaptation.

Fat is an important component of milk and an important source of energy. The fats in milk mainly include triglycerides, diglycerols, monoglycerols, free fatty acids, phosphoesters and sterols [52]. Studies have shown that mastitis enhances the hydrolysis of lipids, so the free fatty acids in milk of dairy cows with mastitis increase [53]. However, the changes in lipid metabolism in this study were mainly manifested in three metabolic pathways, namely, arachidonic acid metabolism, α-linolenic acid metabolism and glycerophospholipid metabolism. No increase in free fatty acids was observed, which might be because free fatty acids are used to provide energy for the immune system to resist the attack of pathogens [54].

Arachidonic acid (ARA) metabolism is important for animal health and tissue homeostasis [55]. ARA is mainly found in the phospholipids of cell membranes and lipid droplets (LDs) of immune cells [56,57]. ARA itself gives cell membranes flexibility and fluidity, acting as a lipid second messenger in cell signaling. In addition, studies have shown that cell membrane mobility is also important for microbial containment [58]. Additionally, recent studies have shown that various bioactive metabolites produced by ARA are closely related to lipid metabolism and immune function [59], which indicates that ARA metabolism has an inevitable link between lipid metabolism and immunity [60]. This is consistent with the results of differential regulation of ARA metabolism found in this experiment in mastitis milk lipidomics screening, which may be achieved through the inflammatory reaction activated by tumor bad factor receptor and Toll-like receptor 4 [61].

α-linolenic acid metabolism is a type of lipid metabolism, whose substrate α-linolenic acid and linoleic acid belong to the polyunsaturated fatty acids (PUFA) of omega-3 and ω-6, respectively. Studies have shown that alpha-linolenic acid can alter cell membrane structure and fluidity, signal transduction pathways such as protein kinases and expression of inflammatory genes [62]. The metabolism of α-linolenic acid and linoleic acid is enhanced after inflammatory stimulation of BMECs [63], which is consistent with the results of this study. It may be that the downstream lipid mediators such as PGG2, 15-keto-PGE2 and 11-HETE, which are related to proinflammatory and anti-inflammatory inflammation, are involved in the inflammatory response to cope with mammary inflammation [64].

Glycerophospholipids can be divided into different types based on the different substituents. The most prominent glycerophospholipids are phosphatidyl choline (PCs), phosphatidyl ethanolamine (PEs), phosphatidyl acid (PAs) and phosphatidyl inositol (PI), which can be used as substrate sources for the production of various lysophospholipids, such as PC, to form LysoPCs [65]. The disorder of glycerophospholipid metabolism can reflect systemic changes caused by inflammatory response. Studies have shown that PC metabolism is enhanced under inflammatory conditions, which is mainly achieved through the action of phospholipase [66]. Inflammatory cytokines can induce the transport of phospholipase A2 (PLA2) enzymes to cell membranes containing glycerophospholipids. Some glycerophospholipids are activated and converted into PUFAs, producing oxylipins that promote the inflammatory cycle. The reduced levels of glycerophospholipids (PC, PE, PS and PI) in mastitis cows may be related to their metabolism to produce oxylipids [67]. Although we did not find any oxylipids in this study, we did identify several important precursors in this pathway, including LA, arachidonic acid and PC [68]. In addition, elevated levels of lysophospholipids, especially LPC, in mastitis bovine milk suggest PLA2 activation, which is consistent with the observed persistent inflammatory process [69]. Our data suggest that metabolites in lipid metabolism can be used as metabolic markers of mastitis.

## 5. Conclusions

To sum up, we found that the microflora in milk after mastitis infection was disordered, *Bifidobacterium, Ureaplasma, Atopococcus, Mycoplasma, Psychrobacter, Ruminobacter, Trepome* may play an important role in regulating dairy cows’ milk quality. In addition, the organisms can adapt to mastitis by regulating arachidonic acid metabolism, α-linolenic acid metabolism and glycerophospholipid metabolism. The results showed that mastitis may cause abnormal lipid metabolism in milk by regulating the diversity of milk microflora, and ultimately affect the milk quality.

## Figures and Tables

**Figure 1 animals-13-02773-f001:**
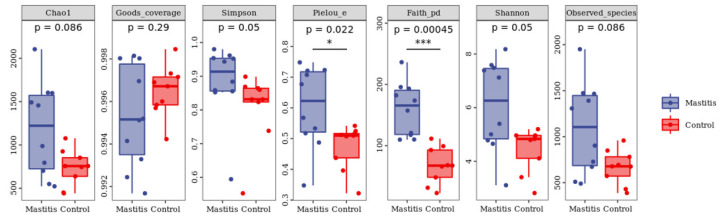
Effects of mastitis on alpha diversity in the milk. The number under the diversity index label is the *p*-value of the Kruskal–Wallis test. * represents the difference is significant; *** represents the difference is extremely significant.

**Figure 2 animals-13-02773-f002:**
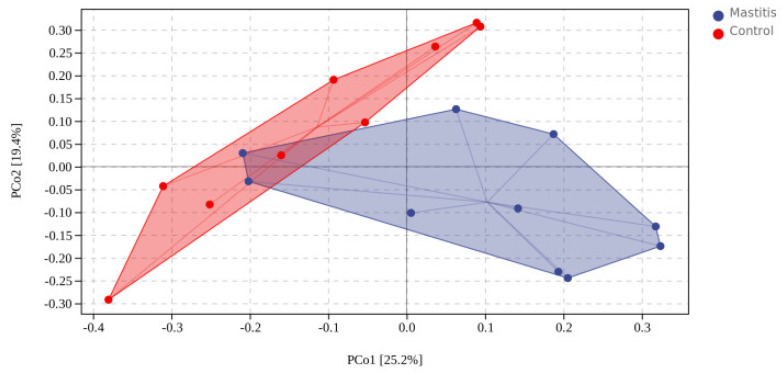
Distance matrix and PCoA analysis. The PCoA plot was generated using unweighted UniFrac−based.

**Figure 3 animals-13-02773-f003:**
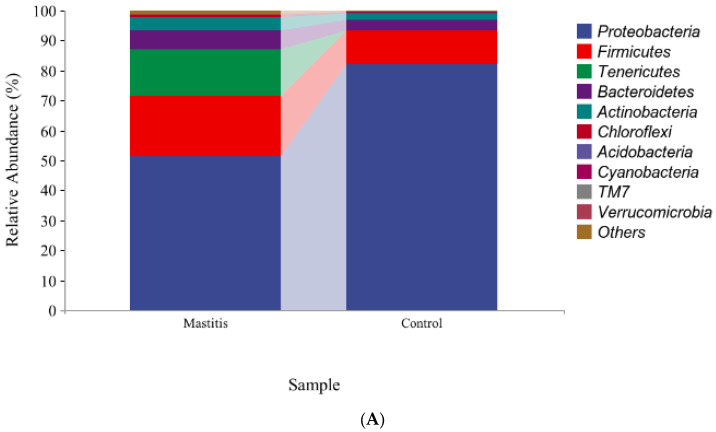
Relative abundance of bacteria at the phylum level in the milk (**A**). Comparison of relative abundances at phylum levels (**B**–**E**) were analyzed using the Kruskal–Wallis rank-sum test. Values are expressed as means ± SEM indicated by vertical bars. * Significantly different means (*p* < 0.05).

**Figure 4 animals-13-02773-f004:**
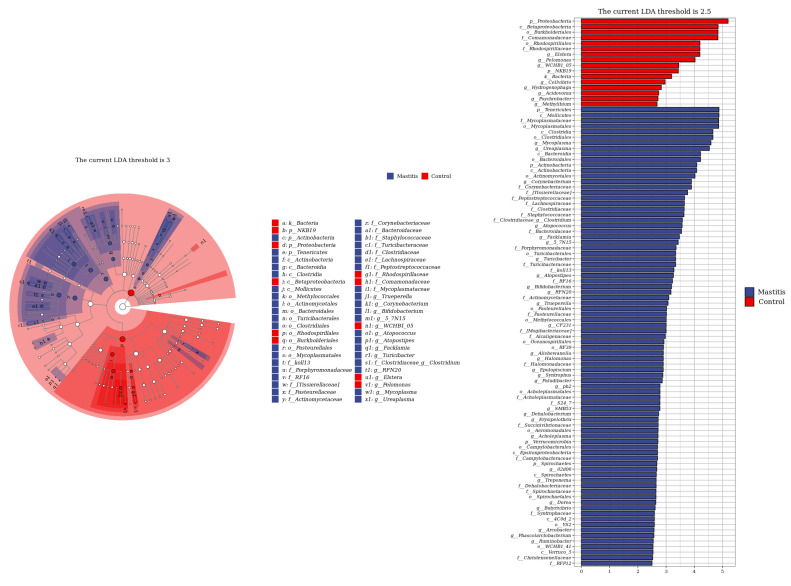
Linear discriminant analysis (LDA) effect size (LEfSe) analysis identified the most differentially abundant from phylum to genus. The genus with linear discriminant analysis values higher than 2.5 is displayed. The length of the bar column represents the LDA score. The cladogram, circles radiating from inner side to outer side, represents the differences in the relative abundance of taxa from phylum to genus level between the two groups.

**Figure 5 animals-13-02773-f005:**
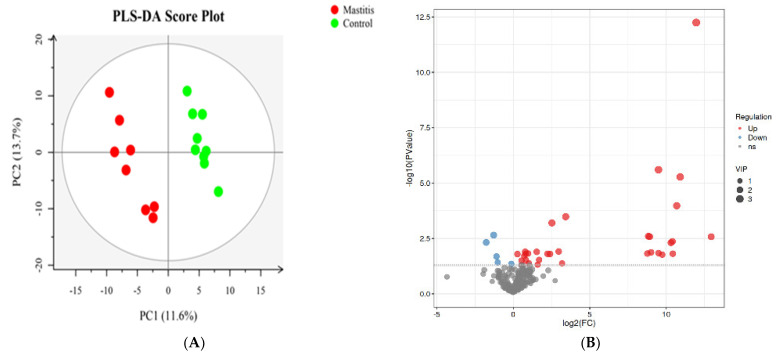
Partial least squares discriminant analysis (PLS-DA). The PLS−DA of microbial metabolites in milk of dairy cows (**A**). Volcano plots showing the differential metabolites (red dot: significantly up-produced metabolites, blue dot: significantly down-produced metabolites, gray dot: the metabolites with no significant difference, Q < 0.05) (**B**).

**Figure 6 animals-13-02773-f006:**
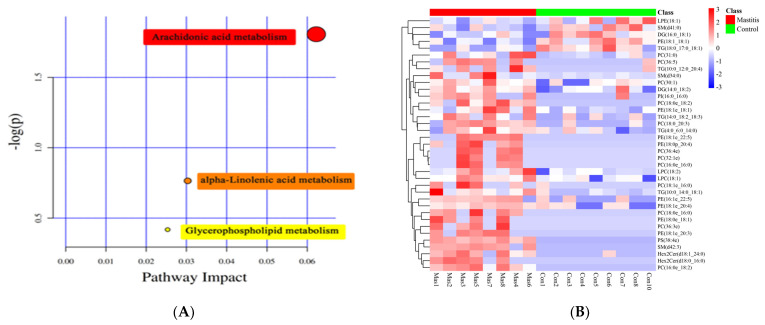
KEGG classification of differentially accumulated MS2 metabolites in the Mas and Con group. The color indicates the size of the *p*−value, and the smaller the *p*-value, the redder the color, and the more significant the enrichment degree (**A**). Heat-map of hierarchical clustering analysis for differential metabolites in Mastitis and Con groups. The color blocks at different positions represent the correlation coefficients between metabolites at corresponding positions. Red represents positive correlation, blue represents negative correlation, and the darker the color, the stronger the correlation (**B**).

**Figure 7 animals-13-02773-f007:**
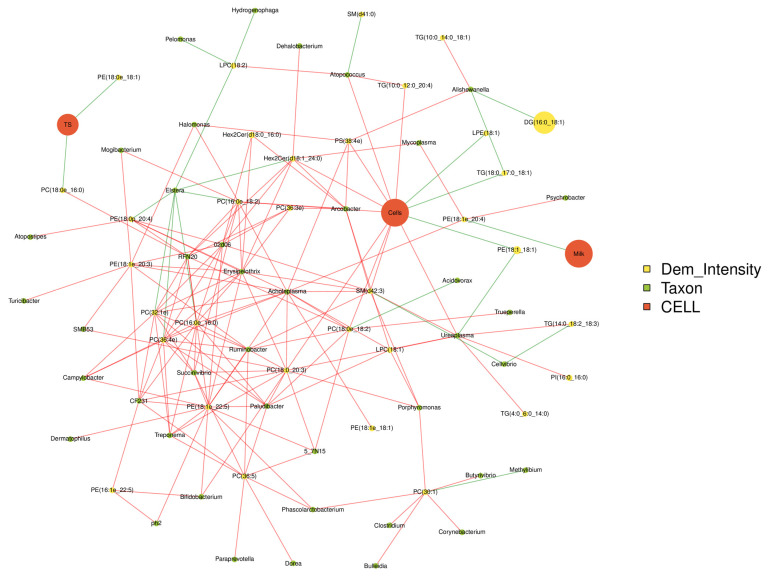
Correlations network analysis among milk qualities, the relative abundance of predominant bacteria at the genus level and differential lipids in the milk. The red line represented positive correlation and the green link represented negative correlation.

**Table 1 animals-13-02773-t001:** Ingredient composition and nutrition levels of the diet (% of DM) [17].

Items	Content (%)	Reference
Diet composition		[17]
Chinese leymus	37.5	
Corn silage	22.5	
Corn	15.2	
Wheat bran	5.3	
Soybean meal	9.2	
DDGS	8.4	
Calcium hydrophosphate	1.4	
Premix1	0.5	
Nutrient composition		
CP	13.1	
NDF	39.6	
Ca	0.6	
P	0.4	
NEL2, MJ/kg DM	5.4	

Note: 1 one kilogram of premix contained mixed vitamins, 800,000 IU; Fe, 1500 mg; Cu, 1000 mg; Zn, 11,000 mg; Mn, 3500 mg; Se, 80 mg; I, 200 mg; and Co, 50 mg. Additionally, 2 NEL was calculated [17].

**Table 2 animals-13-02773-t002:** Effects of mastitis on milk performance of Holstein dairy cows.

Item	Con	Mas	P
Milk production (kg)	18.54 ± 1.290	15.46 ± 1.629	0.058
Milk fat (%)	3.74 ± 0.293	3.35 ± 0.351	0.217
Milk protein (%)	3.57 ± 0.146	3.57 ± 0.509	0.986
Lactose (%)	5.00 ± 0.136	4.71 ± 0.519	0.118
Total solids (%)	12.88 ± 0.733	12.23 ± 1.658	0.271
SCC (1000/mL)	21.25 ± 8.74	8380.0 ± 1885.96	0.001
Urea nitrogen (mg/dL)	25.11 ± 6.45	26.98 ± 9.39	0.610
DIM(d)	308.30 ± 3.529	308.80 ± 2.781	0.729

Note: Con—control; Mas—mastitis; SCC—somatic cell count; DIM—Days in Milk.

## Data Availability

The original contributions presented in the study are included in the article/Appendix A; further inquiries can be directed to the corresponding authors.

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
