# Peer review of "Relationship between Microflora Changes and Mammary Lipid Metabolism in Dairy Cows with Mastitis"

_animals, 2023, doi:10.3390/ani13172773_

Round 1
Reviewer 1 Report
Some revision is suggested to improve the manuscript.
1. Please add line numbers to the full text.
2. Line96-98 Is this part of the description correct? Please check and modify. For the milk samples collected here, do you test them directly with fresh milk or after freezing?
3. Line126 How the lipidome analysis was carried out, please explain clearly.
4. The resolution of the Figure 4 is not high enough. Is there a higher resolution picture?
5. Line 261 Is Spearman's correlation analysis done at the company or by yourself? How do you do it?
6. Line 308-310 Studies have confirmed that Tenericutes were related to colitis in the digestive tract.” The cited article finishes with the following sentence “it is the opinion of the author that the existence of an intramammary microbiota is a fiction that could cause confusion and interfere with practices that have proved useful for mastitis control [34]. Probably the authors found an increase of Tenericutes (that live in the environment) because Mycoplasma spp. (from Tenericutes) was responsible of mastitis, I suppose.
The langue in the manuscript should be improved
Author Response
Animals- 2506768: Relationship between microflora changes and mammary lipid metabolism in dairy cows with mastitis.
Response:
Thank you for the comments proposed by both reviewers. When I revised the manuscript according to the proposed comments, I found that the professional knowledge and writing proficiency had been improved. This is very important for the revision and the next writing.
Thank you once again.
The responses for reviewer 1 is as follow:
Reviewer 1:
- Please add line numbers to the full text.
AU: we have revised in the paper.
- Line96-98 Is this part of the description correct? Please check and modify. For the milk samples collected here, do you test them directly with fresh milk or after freezing?
AU: we have revised in the paper, and we test them with fresh milk.
- Line126 How the lipidome analysis was carried out, please explain clearly.
AU: we have supplied this supplementary information in the revised manuscript.
- The resolution of the Figure 4 is not high enough. Is there a higher resolution picture?
AU: we have revised in the paper.
- Line 261 Is Spearman's correlation analysis done at the company or by yourself? How do you do it?
AU: we did at the company, R-cor package was used to calculate spearman correlation, and the relationships among samples were analyzed according to different correlation coefficients. The relevant information |rho| > 0.6 and P value < 0.01 were screened to construct an association network, in which the red line represented positive correlation and the green link represented negative correlation.
- Line 308-310 Studies have confirmed that Tenericutes were related to colitis in the digestive tract.” The cited article finishes with the following sentence “it is the opinion of the author that the existence of an intramammary microbiota is a fiction that could cause confusion and interfere with practices that have proved useful for mastitis control [34]. Probably the authors found an increase of Tenericutes (that live in the environment) because Mycoplasma spp. (from Tenericutes) was responsible of mastitis, I suppose.
AU: Thank you very much for your recognition. Yes, that's our guess, but the final conclusion needs further verification.

Reviewer 2 Report
The paper is interesting but I think that the results are inflated by the low numbers of animals. I have some concerns about it, which I reported in the attached file.

Minor editing are required.
Author Response
Animals- 2506768: Relationship between microflora changes and mammary lipid metabolism in dairy cows with mastitis.
Response:
Thank you for the comments proposed by both reviewers. When I revised the manuscript according to the proposed comments, I found that the professional knowledge and writing proficiency had been improved. This is very important for the revision and the next writing.
Thank you once again.
The response for reviewer 2 is as follow:
Reviewer: 2
- Lines 31-32 the results do not support this sentence. Please rewrite.
AU: we have rewritten in the revised manuscript.
- Line 68. Which studies? please insert references.
AU: we have revised in the paper
- Table 1. NEL. How was calculated? Please clarify or insert reference in the Note.
AU: The formula is shown as below: NEL(Mcal/kg DM)= 0.5501×DE (Mcal/kg DM) -0.0946, DE represents Digestible Energy; DM represents Dry Matter. This method is from the reference“Yanglian Feng, Jianmin Zhou, Xiaoming Zhang, et al. Study on the calculation method of net energy value of feed milk production of dairy cows in China. Chinese Journal of Animal Science,1987(01):8-11.
- Line 96. How many cows? All or only the 20 selected?
AU: When we were sampling, there were one or two hundred dairy cows in milk, and we just randomly sampled 10 of them as normal, and there were 15 cows with mastitis, and we randomly sampled 10 of them.
- Lines 102-106. Could you explain better how you performed the extraction? please insert the ml of milk used. Did you eliminate fat before extraction? How much DNA did you used for the sequencing. Give more detail in general. The pargrph is too sintetic.
AU: 50 mL sample of milk was concentrated for 12min at 3,000 x g (4°C). The precipitates were washed 3 times in a 0.9% NaCl solution. Bacterial DNA was extracted according to the manufacturer's instructions(QIAamp DNA Stool Mini Kit,Qiagen,Valencia, USA). The quality of metagenomics DNA was confirmed by 1% agarose gel electrophoresis and diluted to 1 ng/uL as the template for PCR.
- Lines 102-106 Which hypervariable regions of the 16S rRNA gene did you amplified? Please clarify.
AU: The V3 and V4 regions of 16S rDNA genes were amplified using the universal forward primer 338F (5'-ACTCCTACGGGAGgCAGCA-3') and 806R (5'-GGactachVGGGTWTCTAAT-3') with a set of 6-nucleotide barcodes..
- Lines 111-113. please insert references related to the softwares used.
AU: we have revised in the manuscript, and provided a reference.
- Lines 114-119.Which methods? please clarify! What does it mean "a variety of .....". Which statistical test????
AU: The alpha diversity of the milk bacteria was estimated using the number of ASV,Chao1,Shannon indices,Simpson, Pielou, Faith Observed species and Good's coverage implemented in QIIME2. Kruskal-Wallis test was used to test whether a significant difference was existed between the alpha diversity of two groups. The beta diversity of samples were estimated using Unweighted UniFrac. The difference of beta diversity between two groups were tested by analysis of similarity (ANOSIM). Greengenes database() was used to annotate species taxonomy, construct association network, calculate topological index, and try to find the key species.
- Lines 126. 300 ????? please insert the right unit. In many sentences there are errors in units.
AU: we have revised in the paper.
- Lipid search software. Please add reference or explain better.
AU: The LipidSearch software database contains more than two molecular lipid species (96 sub-classes) and their predicted fragment ions and provides the most complete coverage of different sample types including cells, human plasma, insect larvae, plants, seed oils, tissue and yeast.
- Table 2. DIM is Days in milk not dairy in milk!!!
AU: Sorry, we made a mistake here, we have revised in the paper.
- Fig 4. The figure is not clear. please provide an image at higher resolution.
AU: we have provided a higher resolution picture in the revised manuscript.
- the figure 5 B is a volcano plot not an heat Map. In the figure there are blue, red and gray dot not green! Please check!
AU: Sorry, we made a mistake here, we have revised in the paper.
- Fig 6 The caption is not clear. Please add more details.
AU: we have revised in the paper.
- Fig 7. Please use an image at higher quality.
AU: we have revised in the paper.
- Lines 272-274. In your experiment the difference in milk yield between the two groups are not significant. P=0.06 is >0.05. Probably the number of sample is too small to obtain significant results.
AU: The question you raised is very interesting and very important. The control group was randomly selected from non-mastitis cows (about 100 to 200 dairy cows), and the experimental group was randomly selected from all mastitis animals (about 15 dairy cows). Moreover, in the summer in Changsha, heat stress also has a certain impact on milk production of cows, so the difference in milk production is not significant. This is our speculation, the specific reasons need to be further explored.
- Lines 280-281. In your experiment the differences are not significant!!!!! Please formulate your comments in a way that is consistent with your results.
Lactose (%) Control 5.00±0.136 Mast 4.71±0.519 P=0.118
AU: Thank you for your valuable advice. The fact is that the lactose content, according to the specific data, there is a certain degree of reduction, but it has not reached a significant reduction. We have revised the description of this part in the manuscript.
- Lines 284-287 this statement is based on what outcome. Please clarify.
AU: Thank you for your reminding. The explanation we provided before may not be consistent with the description here, and we have revised it in the manuscript.

Round 2
Reviewer 2 Report
Dear authors,
I found your paper slightly improved. Hovewer some yours reply to my comments were not added to the text and are confused!
1An example is your replay to my comments n.3 about NED. You don't have only replay to me, you have to improve the text and add the information that you gave me.
2 About the DNA extraction: have you user CTAB or this kit? 50 mL sample of milk was concentrated for 12min at 3,000 x g (4°C). The precipitates were washed 3 times in a 0.9% NaCl solution. Bacterial DNA was extracted according to the manufacturer's instructions(QIAamp DNA Stool Mini Kit,Qiagen,Valencia, USA). The quality of metagenomics DNA was confirmed by 1% agarose gel electrophoresis and diluted to 1 ng/uL as the template for PCR. I THINK YOU ARE REALLY CONFUSED IF YOU DON'T KNOW WHICH METHOD HAVE YOU USED. Please insert in the text the right method.
3 Please insert in the text the information about the amplified regions.
The V3 and V4 regions of 16S rDNA genes were amplified.
English is good
Author Response
Animals- 2506768: Relationship between microflora changes and mammary lipid metabolism in dairy cows with mastitis.
Response:
Thank you for the comments proposed by both reviewers. When I revised the manuscript according to the proposed comments, I found that the professional knowledge and writing proficiency had been improved. This is very important for the revision and the next writing.
Thank you once again.
The response for reviewer 2 is as follow:
- An example is your replay to my comments n.3 about NED. You don't have only replay to me, you have to improve the text and add the information that you gave me.
AU: Thank you for your reminder, we have revised in the paper according to your suggestion.
- About the DNA extraction: have you user CTAB or this kit? 50 mL sample of milk was concentrated for 12min at 3,000 x g (4°C). The precipitates were washed 3 times in a 0.9% NaCl solution. Bacterial DNA was extracted according to the manufacturer's instructions(QIAamp DNA Stool Mini Kit, Qiagen, Valencia, USA). The quality of metagenomics DNA was confirmed by 1% agarose gel electrophoresis and diluted to 1 ng/uL as the template for PCR. I THINK YOU ARE REALLY CONFUSED IF YOU DON'T KNOW WHICH METHOD HAVE YOU USED. Please insert in the text the right method.
AU: In total, 10 mL milk sample was concentrated for 12min at 3,000 x g (4°C). The precipitates were washed 3 times in a 0.9% NaCl solution. Bacterial DNA was extracted according to the manufacturer's instructions (QIAamp DNA Stool Mini Kit, Qiagen, Valencia, USA). The quality of metagenomics DNA was confirmed by 1% agarose gel electrophoresis and diluted to 1 ng/μl as the template for PCR. And we have revised in the manuscript.
- Please insert in the text the information about the amplified regions.
The V3 and V4 regions of 16S rDNA genes were amplified.
AU: The V3 and V4 regions of 16S rDNA genes were amplified using the universal forward primer 338F (5'-ACTCCTACGGGAGGCAGCA-3') and 806R (5'-GGACTACHVGGGTWTCTAAT-3') with a set of 6-nucleotide barcodes, and we have revised in the manuscript.